# Bioinformatics analysis identifies DYNC1I1 as prognosis marker in male patients with liver hepatocellular carcinoma

Jian Zhou[1☯], Yue Zhu[2☯], Songlin Ma[3☯], Yi Li[1], Kun Liu[4], Sihuan Xu[2], Xin Li[2], Li Li[5], Junfang Hu[6], Yan Liu[2]*

1 Department of Infectious Diseases, Puren Hospital Affiliated to Wuhan University of Science and Technology, Wuhan, Hubei, China, 2 Biological Cell Therapy Research Center, Puren Hospital Affiliated to Wuhan University of Science and Technology, Wuhan, Hubei, China, 3 Department of Gastroenterology, The Central Hospital of Wuhan, Tongji Medical College, Huazhong University of Science and Technology, Wuhan, China, 4 Otorhinolaryngology, Puren Hospital Affiliated to Wuhan University of Science and Technology, Wuhan, Hubei, China, 5 The Ministry of Science and Education, Puren Hospital Affiliated to Wuhan University of Science and Technology, Wuhan, Hubei, China, 6 Department of Pharmacy, Puren Hospital Affiliated to Wuhan University of Science and Technology, Wuhan, Hubei, China

☯ These authors contributed equally to this work.
* liuyan@wust.edu.cn

## Abstract

### Background

Liver hepatocellular carcinoma (LIHC) is one of the most common malignant tumors. However, the etiology and exact molecular mechanism of LIHC are still not fully understood, which makes it urgent for us to further study the molecular events behind.

### Methods

In this study, differences in mRNA expression between LIHC samples and normal adjacent samples were found through analyzing the TCGA database, and key targets were sought. We analyzed 371 LIHC samples and 50 normal adjacent samples according to P <0.01 and logFC>2.5, a total of 1092 genes were identified differentially expressed, including 995 up-regulated genes and 97 down-regulated genes. We predicted the interactions of these differentially expressed mRNAs, and used Cyto-Hubba to locate the hub gene-dynein cytoplasmic 1 intermediate chain 1 (DYNC1I1).

### Results

Survival analysis showed that DYNC1I1 was a prognostic factor for LIHC male patients. Functional enrichment indicated that DYNC1I1 and differentially expressed interacting proteins were involved in the cell cycle.

### Conclusion

In conclusion, this study discovers that DYNC1I1 can be used as a prognostic marker for LIHC male patients.

**Data Availability Statement:** In this study the original sequence data and clinical information can be downloaded from TCGA database (https://

cancergenome.nih.gov/), and all the data generated by the analysis are included in this article.

**Funding:** This study was funded by the Hubei Province health and family planning scientific research project (WJ2019M257) and Wuhan Municipal Health scientific research project (WX20C13). The funders had no role in study design, data collection and interpretation, or the decision to submit the work for publication.

**Competing interests:** The authors have declared that no competing interests exist.

## Introduction

Liver hepatocellular carcinoma (LIHC) is one of the most common malignant tumors of the digestive tract. Globally, the incidence of LIHC ranks the sixth in the incidence of malignant tumors and the fourth in mortality [1]. LIHC seriously affects the lives and health of people. At present, the overall prognosis of LIHC is unsatisfactory. The main reasons include the insidious disease, high degree of malignancy, recurrence and metastasis [2]. Therefore, the identification of LIHC-specific biomarkers can help predict and monitor disease progression, and more importantly, through the implementation of early intervention, cases that may evolve into aggressive diseases can be reduced [3].

The Cancer Genome Atlas Project (TCGA) was jointly launched in 2006 by the National Cancer Institute (NCI) and the National Human Genome Research Institute (NHGRI). The TCGA database contains genome data for 33 tumor projects and provides original sequencing data to all researchers [4]. TCGA has released many mRNA sequencing data of LIHC cancer patients. This study aimed to determine the mRNA expression differences between LIHC samples and normal adjacent samples through analyzing high-throughput mRNA data downloaded from the TCGA database. We used protein interactions [5] and Cyto-Hubba [6] to find the hub gene-DYNC1I1. Also, we assessed the prognostic value of DYNC1I1 and analyzed the possible biological functions of DYNC1I1, which were expected to provide new insights into the underlying molecular mechanisms of LIHC.

## Material and methods

### Data processing

Raw sequencing data and clinical information were downloaded from the TCGA database (https://cancergenome.nih.gov/). Altogether, a total number of 421 samples were enrolled in this study, including 371 LIHC samples and 50 normal adjacent samples. We used the R language package to process mRNAs sequencing data. The differentially expressed mRNAs between LIHC samples and normal adjacent samples are analyzed by the "DESeq2" package in R. When we were calculating the fold change (FC) of individual mRNA expression, we considered that the differentially expressed mRNAs with $P < 0.01$ and $logFC > 2.5$ as significant.

### Differential gene analysis

The R package of "DESeq2_1.28.1" [7] was used for Principal Component Analysis (PCA) and MA-plot of significant differentially expressed mRNA, and we used the code "dds <- dds[row-Sums(counts(dds))>1,]" to delete low-expressed genes. The "pheatmap_1.0.12" package [8] and "ggrepel_0.8.2" package [9] in R were used for heatmap and volcano map, respectively. The default parameters were used for all analysis.

### Functional enrichment analysis

We used the R package "ClusterProfiler_3.16.1" [10] for gene Ontology (GO) enrichment and used the online database DAVID [11] for gene Kyoto Encyclopedia of Genes and Genomics (KEGG) pathway analysis. GO terms or KEGG pathways with $P < 0.05$ were considered statistically significant.

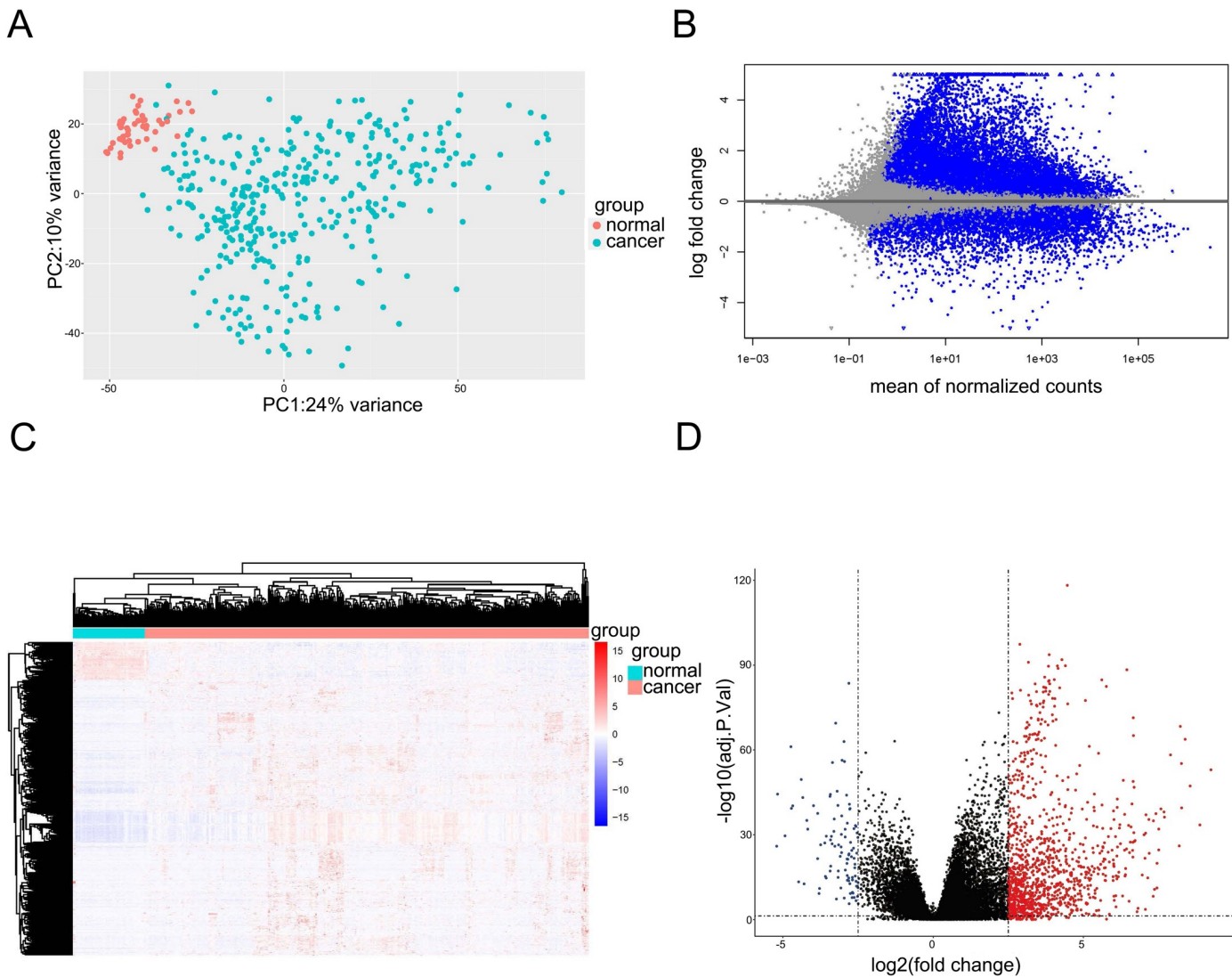

**Fig 1. Identification of expression differences between LIHC samples and normal adjacent samples.** A. PCA of 371 LIHC samples and 50 normal adjacent samples expression analysis. Red represented normal adjacent samples, and blue represented LIHC samples; B.MA-plot of the differential mRNA expression analysis; C. Heat map of the differential mRNA expression analysis. Blue represented normal adjacent samples, and red represented LIHC samples; D. Volcano plot of the differential mRNA expression analysis. Red represented differential mRNAs that were highly expressed in LIHC samples.

## Protein-Protein Interaction (PPI) network construction and analysis of modules

STRING is a search tool that can analyze the interaction relationship between genes/proteins (https://string-db.org/). Using STRING to analyze the PPI network can help us understand the relationships between different genes/proteins. Cyto-Hubba software was used to screen for hub genes. We used 11 algorithms of the Cyto-Hubba software to screen out the hub gene: MCC, DMNC, MNC, Degree, EPC, BottleNeck, EcCentricity, Closeness, Radiality, Betweenness, Stress.

## Prognosis analysis

We used the LIHC patient data in TCGA and the "Survival" package and "Survminer" package of R language to draw the survival curve of LIHC patients.

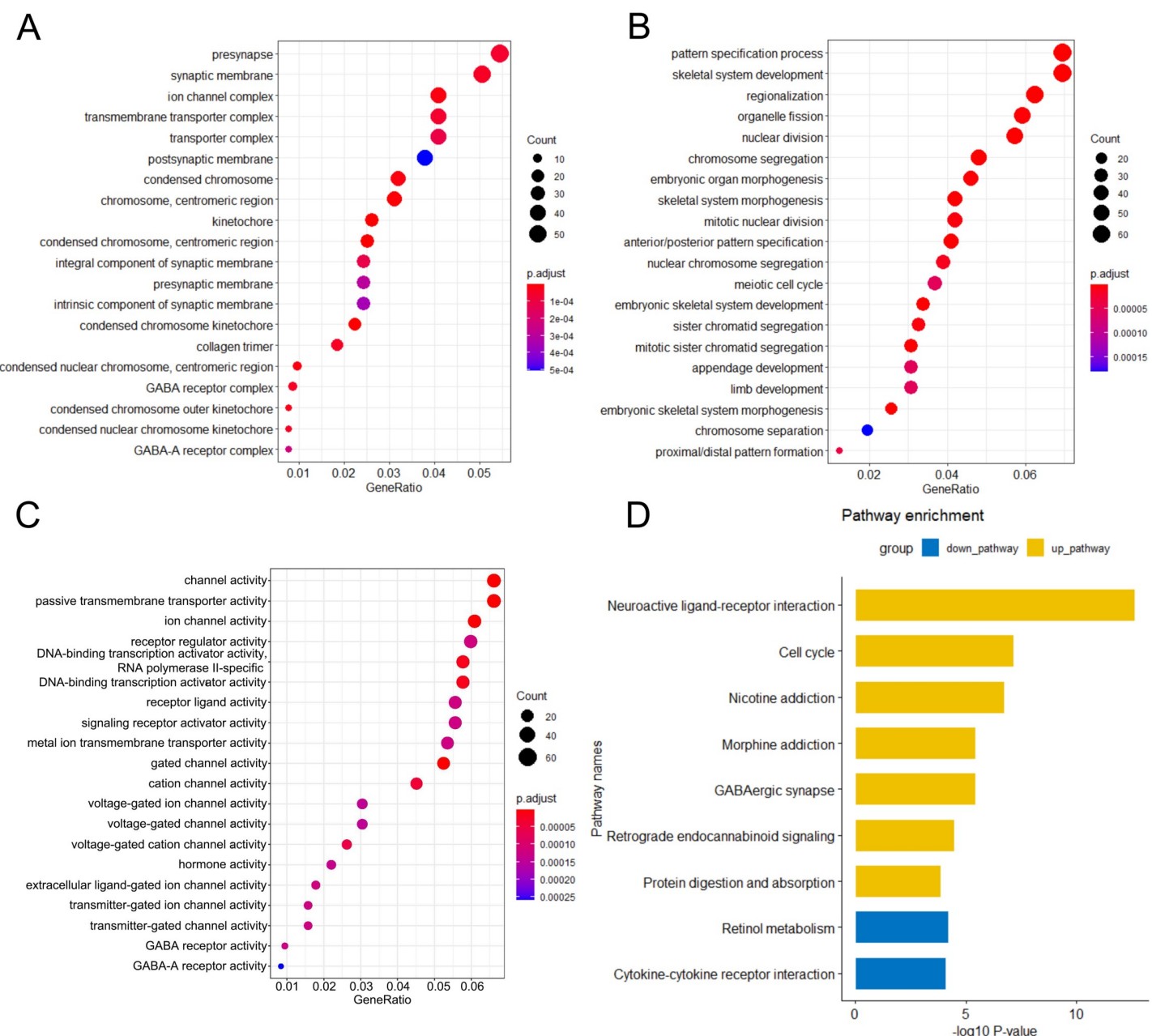

**Fig 2. Functional enrichment analysis of differentially expressed mRNAs.** The most important GO terms for cell component (A), biological process (B) and molecular function (C) of 1092 differential genes were shown in the figures; D. Gene networks identified through KEGG pathway analysis of the differentially expressed mRNAs.

## Results

### Identification of differentially expressed mRNAs in LIHC

We retrospectively analyzed the TCGA LIHC dataset. In the present study, a total of 421 samples were enrolled in this study, including 371 LIHC samples and 50 normal adjacent samples, to identify the differentially expressed genes. PCA showed that LIHC samples and normal adjacent samples fell in different areas, and the two groups of samples were effectively

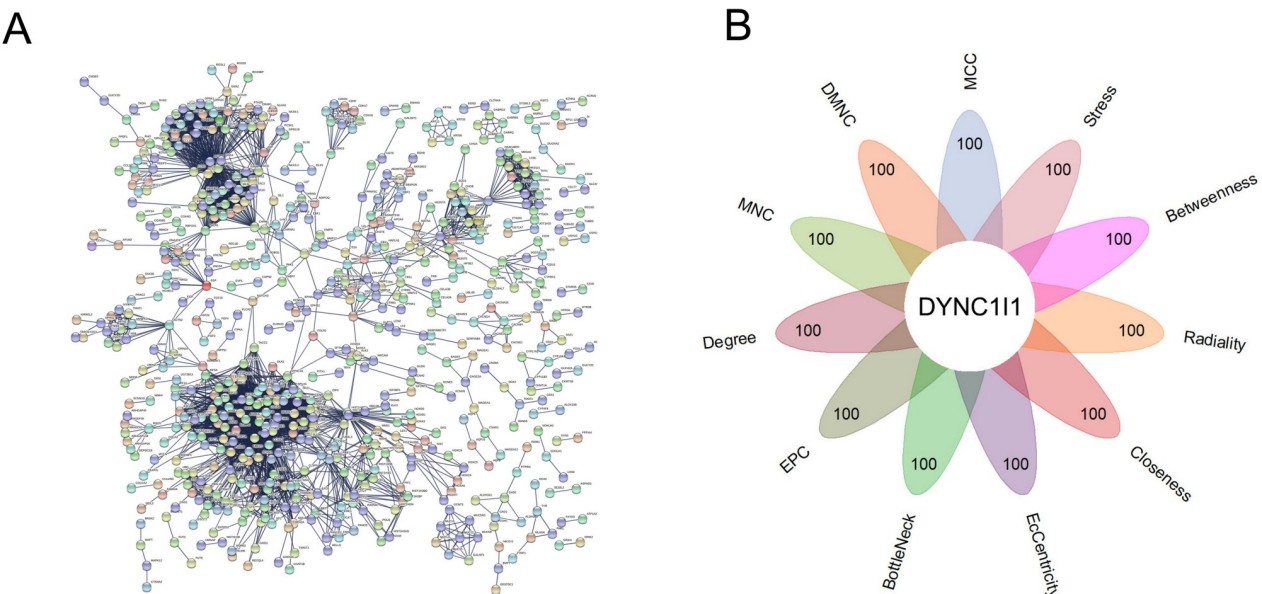

**Fig 3. Identify the hub gene.** A. Use STRING online database to build a PPI network that interacts with DYNC1I1; B. Use Cyto-Hubba software to identify the hub gene. DYNC1I1 was the only hub gene that intersects among the top 100 genes obtained by each of the 11 algorithms.

distinguished (Fig 1A, 1B) depicted the transcription abundance values of LIHC samples and normal adjacent samples as MA-plot. According to P < 0.01 and logFC > 2.5, a total number of 1092 differentially expressed mRNAs were identified between LIHC samples and normal adjacent samples, including 995 up-regulated and 97 down-regulated mRNAs. We presented the result in the form of Heat map (Fig 1C) and Volcano plot (Fig 1D).

## Functional annotation of differentially expressed mRNAs in LIHC

In order to understand the biological roles of the 1092 differentially expressed mRNAs in LIHC, we performed GO and KEGG pathway enrichment analysis using the R package "Clusterprofiler", which showed 20 most significant cellular components, biological process and molecular function terms, and 9 significant KEGG pathways. GO analysis revealed that major terms enriched in the cellular component terms were ion channel complex and transmembrane transporter complex (Fig 2A). The most significantly enriched biological process category were organelle fission and nuclear division (Fig 2B). For the molecular function category, the primary enriched terms were passive transmembrane transporter activity, ion channel activity, and DNA−binding transcription activator activity (Fig 2C). KEGG pathway analysis indicates that the differentially expressed mRNAs played relevant roles in cell cycle, cytokine−cytokine receptor interaction (Fig 2D). The specific details of GO and KEGG pathway enrichment analysis are in the S1−S4 Tables.

## Using string and Cyto-Hubba to identify the hub gene in LIHC

We used STRING online database to analyze 1092 differentially expressed mRNAs and construct PPI network (Fig 3A), for details about PPI calculation in STRING, see S5 Table. Then, we downloaded the results and use Cyto-Hubba software for analysis. We analyzed the results of PPI using 11 algorithms of Cyto-Hubba software. We extracted the top 100 hub genes from 11 algorithms, and get the only key gene DYNC1I1 after taking the intersection (Fig 3B). The detailed scoring of genes in PPI by 11 algorithms was in the S6 Table.

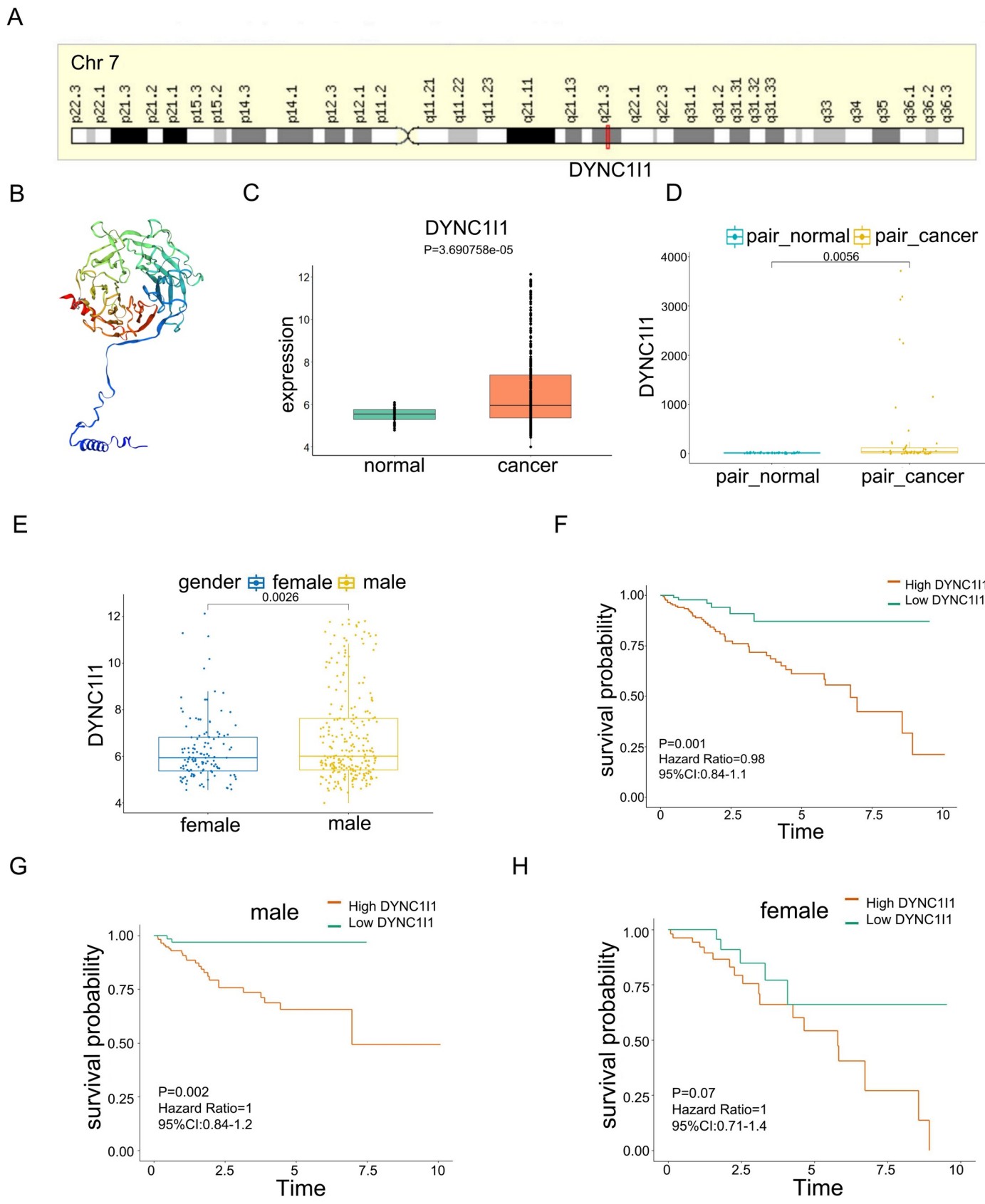

**Fig 4. Characterization of DYNC1I1.** Genomic localization (A) and protein structure (B) of DYNC1I1; C. Differential expression of DYNC1I1 in LIHC samples and normal adjacent samples (P = 3.690758e-05); D. Differential expression of DYNC1I1 in 50 pairs of LIHC samples and normal adjacent samples (P = 0.0056); E. Differential expression of DYNC1I1 in LIHC male patients and female patients (P = 0.0026); F. Association between DYNC1I1 expression and disease-free survival time in the TCGA-LIHC dataset (P = 0.001). G-H. DYNC1I1 had a significant impact on the survival of LIHC male patients (P = 0.002), but has no significant impact on the survival of LIHC female patients (P = 0.07).

## Characterization of DYNC1I1

DYNC1I1 (Dynein Cytoplasmic 1 Intermediate Chain 1) is an important cargo-binding subunit of cytoplasmic dynein. It serves as one of several non-catalytic accessory components of the cytoplasmic dynein 1 complex that are thought to be involved in linking dynein to cargos and to adapter proteins that regulate dynein function. DYNC1I1 is located at chr7: 95772554–96110322 (Fig 4A). We use online database SWISS–MODEL [12] to examine the protein structure of DYNC1I1, Fig 4B showed the protein structure of DYNC1I1, the structure consists of multiple beta folds to form a compact space structure. There was an alpha helix at the N-terminal and the C-terminal, and the alpha helix at the N-terminal is free to the outside. We detected the expression of DYNC1I1 in LIHC data of the TCGA database, and the results showed that the expression of DYNC1I1 in LIHC samples was higher than in normal adjacent samples, and the P value = 3.690758e-05, which was statistically significant (Fig 4C). We extracted 50 pairs of LIHC samples in the TCGA data, and the results showed that the expression of DYNC1I1 in the paired LIHC samples was also higher than in the paired normal adjacent samples, and the P value = 0.0056 (Fig 4D). The expression of DYNC1I1 in LIHC male patients and female patients was also different (P = 0.0026), and the results showed that its expression in male patients was higher than in female patients (Fig 4E). We further explored whether the elevated expression of DYNC1I1 level in LIHC affected the survival of patients. LIHC data with gene expression and clinical information from TCGA were used to investigate DYNC1I1 prognostic significances. The survival rate of DYNC1I1 high expression group was significantly lower (P = 0.001, hazard ratio: 0.98, 95% CI: 0.84–1.1, Fig 4F). Our previous results showed that the expression of DYNC1I1 in LIHC male patients was higher than in female patients. Therefore, we separately investigated the impact of DYNC1I1 on the survival of LIHC male and female patients. The results found that DYNC1I1 had a significant impact on the survival of male patients (Fig 4G), while it had not statistically significant impact on the survival of female patients (Fig 4H). These results indicated that high expression of DYNC1I1 was an adverse factor for the survival of LIHC male patients.

## Function analysis of DYNC1I1

We extracted the proteins that interact with DYNC1I1 in the PPI results (Fig 5A), The specific details of the PPI results were in the Table 1. To reveal the potential biological functions of these interacting proteins, we conducted GO and KEGG analysis. GO function annotation of the interacting proteins was performed using the R package "Clusterprofiler". The most significant GO terms for cellular component, biological process and molecular function were shown in Fig 5B–5D. The KEGG pathway analysis was performed using the DAVID database, and the results of the analysis were shown in Fig 5E. The interacting proteins were mainly enriched in cell cycle, Progesterone–mediated oocyte maturation and Oocyte meiosis. This analysis indicated that DYNC1I1 was mainly involved in cell cycle. The specific details of GO and KEGG pathway enrichment analysis were in the S7–10 Tables.

## Discussion

LIHC is a highly malignant tumor with a high mortality rate. The treatment of LIHC is mainly surgical resection, but due to its high recurrence and metastasis rate, the prognosis of LIHC

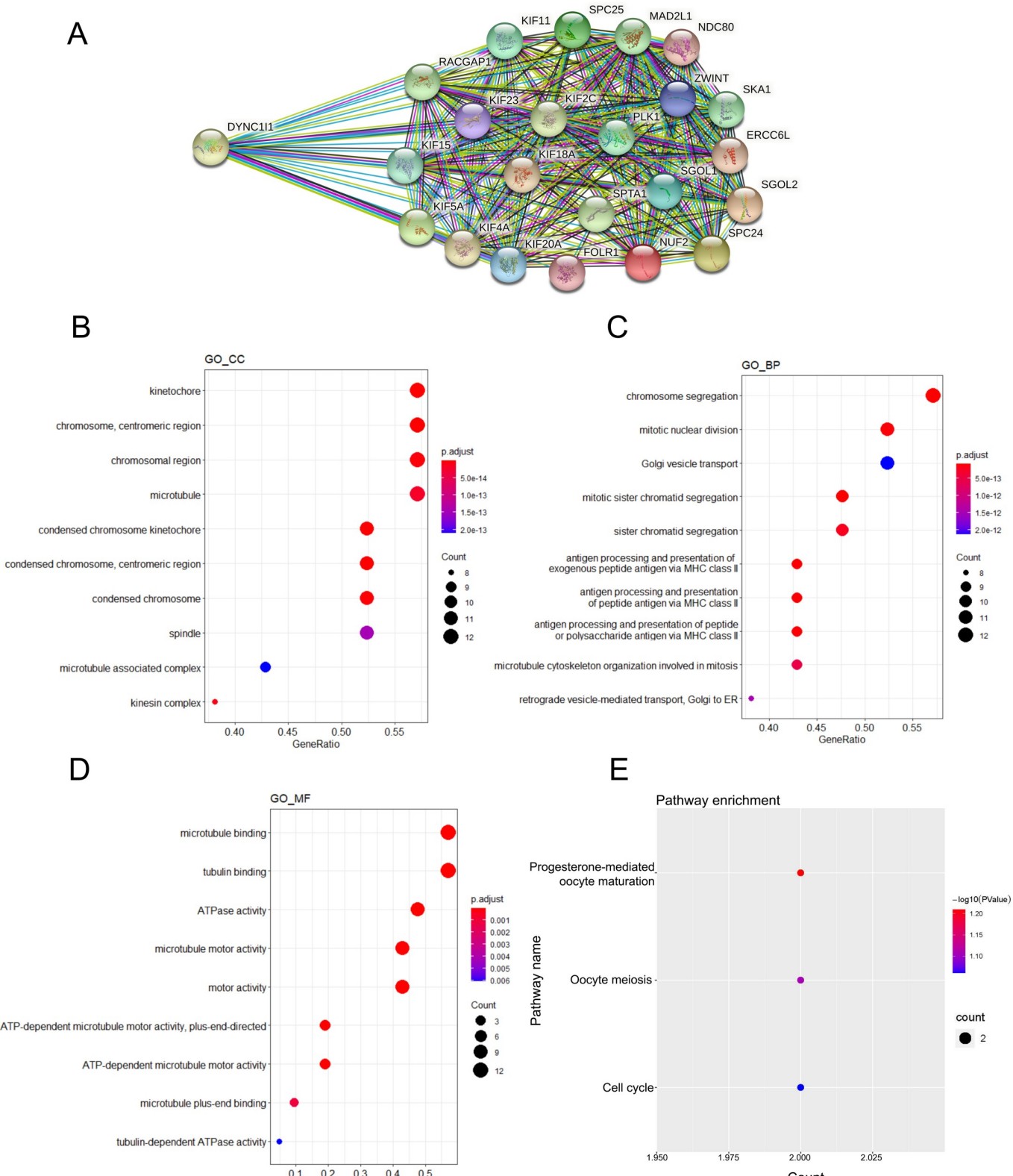

**Fig 5. Function analysis of DYNC1I1.** The figure showed the cellular component (A), biological process (B) and molecular function (C) of DYNC1I1 and the proteins that interacted with it (D); E. Possible pathways mediated by DYNC1I1 and its interacting proteins. The color of the dot represents the P value, and the size of the dot represented the number of counts.

**Table 1. Proteins that interact with DYNC1I1.**

| Gene | PPI |
|------|-----|
| DYNC1I1 | KIF20A KIF11 MAD2L1 ZWINT SPTA1 FOLR1 PLK1 SGOL1 NDC80 NUF2 SKA1 SPC24 KIF4A KIF18A RACGAP1 SGOL2 KIF23 SPC25 KIF15 KIF5A ERCC6L KIF2C |

patients is often far from satisfactory [13, 14]. Due to the insidious onset, rapid progress, high recurrence and metastasis rate of LIHC, the prognosis of patients is poor, the overall 5-year survival remains at 25%-39%, and the recurrence rate of advanced LIHC patients is about 80% [15]. Therefore, a better understanding of the progression of LIHC and new mechanisms of occurrence and progression will help us find new therapeutic targets, formulate more effective treatment strategies, and extend the survival time of LIHC patients.

At present, the first-line targeted drugs for LIHC treatment include levatinib and sorafenib, and the second-line targeted drugs include cabotinib, regorafenib, ramucirumab and nivolumab [16]. In order to improve the prognosis of patients with LIHC and advance development of this field, more researches are needed on molecular targeted therapies that use genome maps and biomarker matching [17, 18]. With the advancement of RNA sequencing and other technologies, the mechanism of the occurrence and progress of LIHC is constantly being explored, and more targets are identified and utilized [19, 20].

In the present study, we find that DYNC1I1 is differentially expressed in LIHC patients, and it is significantly related to patient survival. Previous studies have shown that DYNC1I1 can promote the progression and metastasis of gastric cancer [21, 22], colon cancer [23] and glioblastoma [24], but no study has found that DYNC1I1 has a biological function in LIHC and is a prognostic factor of LIHC. The content of our research makes researchers better understand the biological functions of DYNC1I1.

In order to gain new insight into the molecular functions of DYNC1I1, we screened the differentially expressed genes that interact with it, analyzed related pathways, and performed GO annotations. Abnormal signal pathways play a vital role in the occurrence and development of LIHC in male patients. We discover that DYNC1I1 can regulate the cell cycle and is related to some transmembrane transportation, which indicates that DYNC1I1 may not only affect the progress of LIHC in male patients, but also affect drug transportation.

In summary, we have identified DYNC1I1 as a potential prognostic predictor of LIHC for male patients. In future research, we will classify patients' samples by sex and use a larger sample size to verify our research results, and the molecular mechanism of DYNC1I1 in progression of LIHC also needs more in-depth functional researches.

## Supporting information

**S1 Table.**
(XLSX)

**S2 Table.**
(XLSX)

**S3 Table.**
(XLSX)

**S4 Table.**
(XLSX)

**S5 Table.**
(XLSX)

**S6 Table.**
(XLSX)

**S7 Table.**
(XLSX)

**S8 Table.**
(XLSX)

**S9 Table.**
(XLSX)

**S10 Table.**
(XLSX)

## Author Contributions

**Conceptualization:** Yan Liu.

**Data curation:** Jian Zhou, Yue Zhu, Songlin Ma, Yi Li, Kun Liu, Sihuan Xu, Xin Li, Li Li, Junfang Hu, Yan Liu.

**Formal analysis:** Jian Zhou, Yue Zhu, Songlin Ma, Yan Liu.

**Funding acquisition:** Jian Zhou, Yue Zhu.

**Investigation:** Jian Zhou, Yue Zhu, Songlin Ma, Yi Li, Kun Liu, Sihuan Xu.

**Methodology:** Jian Zhou, Yue Zhu, Songlin Ma, Yi Li, Kun Liu, Sihuan Xu, Xin Li, Li Li, Junfang Hu.

**Project administration:** Yan Liu.

**Resources:** Yi Li, Kun Liu, Sihuan Xu, Yan Liu.

**Supervision:** Yan Liu.

**Validation:** Jian Zhou, Yue Zhu, Songlin Ma, Yi Li, Kun Liu, Sihuan Xu, Xin Li, Li Li, Junfang Hu.

**Writing – original draft:** Jian Zhou, Yue Zhu, Songlin Ma.

**Writing – review & editing:** Yi Li, Kun Liu, Sihuan Xu, Xin Li, Li Li, Junfang Hu, Yan Liu.

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
