## [Decision Letter · Decision Letter 0]

4 Jun 2021

PONE-D-21-13904

Bioinformatics analysis identifies DYNC1I1 as prognosis marker in patients with Liver hepatocellular carcinoma

PLOS ONE

Dear Dr. Liu,

Thank you for submitting your manuscript to PLOS ONE. After careful consideration, we feel that it has merit but does not fully meet PLOS ONE’s publication criteria as it currently stands. Therefore, we invite you to submit a revised version of the manuscript that addresses the points raised during the review process.

The reviewers have included different suggestions and have formulated several questions that must be duly answered before the article can be published.

In addition to their questions, I myself have included several questions at the end of this email. Please include an answer to all of them.

We look forward to receiving your revised manuscript.

Kind regards,

Eduardo Andrés-León

Academic Editor

PLOS ONE

Journal Requirements:

Additional Editor Comments:

In the analysis it is stated that 371 samples of LIHC and adjacent normal tissue are used. However, the total number of "normal" samples is 50, so I would like to know if in a paired analysis (among the 50 paired samples) DYNC1I1 still appears as a prominent gene.

On the other hand, the text included in the differential expression analysis is not very detailed. Has any kind of filter been done to eliminate low expressed genes ? with what cpm value ? The P value is adjusted ? with which method ?

Please note that any kind of analysis must include the necessary details for the work to be reproducible by others.

Reviewers' comments:

Reviewer's Responses to Questions

**Comments to the Author**

1. Is the manuscript technically sound, and do the data support the conclusions?

Reviewer #1: Partly

Reviewer #2: Partly

2. Has the statistical analysis been performed appropriately and rigorously? 

Reviewer #1: I Don't Know

Reviewer #2: Yes

3. Have the authors made all data underlying the findings in their manuscript fully available?

Reviewer #1: No

Reviewer #2: Yes

4. Is the manuscript presented in an intelligible fashion and written in standard English?

Reviewer #1: Yes

Reviewer #2: Yes

5. Review Comments to the Author

Reviewer #1: The present work describes the discovery of a novel prognostic biomarker for liver hepatocellular carcinoma patients. Using TCGA available RNA-seq and clinical patients’ data and applying bioinformatics approaches, authors unveil DYNC1|1 as the most significantly deregulated gene in a series of LIHC patients. I would like to address the following aspects related to this manuscript:

Major comments:

1. The main figures of the paper should represent the results described in the text. The figures selected do not have enough quality to be evaluated and the description in the legend is not well described. Please, add high quality figure that could be evaluated, a main title and a complete description of the plots and graphics represented, included the meaning of colour keys (colours should be correlating between figures) and symbols.

2. Line 102-103. Please, include a table (in supplementary data) with the 20 most significant GO terms described in the text and 9 KEGG significant pathways with their score. Provide plots with more quality.

3. Line 119. Please provide a table (supplementary data) with the top ten genes obtained with the 11 different algorithms and their scores. Is only DYNC1|1 in the intersection of these algorithms? If not, include a ranking of the top ten (maximum) and their scores.

4. It would be interesting if authors could provide a table with the interacting genes belonging to the same hub than DYNC1|1. It could complete the information and add some valuable insight about the possible functionality of this alteration in liver cancer (main text).

5. Line 124. In the characterization of DYNC1|1 there is a lack of description. Authors showed in a figure the gene location and protein structure, but they did not add much detail in the text. Please, complete the description (family of the gene, which kind of protein codifies…), features of the structure and potential functionality of the gene (if it was previously described).

6. Line 126. Authors described the significant difference of DYNC1|1 expression between LIHC and normal adjacent tissue, but they did not include the statistical value for this comparison in the text. The same was done when they compared male vs female in LIHC. Please, include all these statistics, and add in the “Methods” sections how these statistics were performed.

7. In line with the previous comments and comparisons performed, the individual comparisons “LIHC vs control” in male and women independently were missed, and due to the observation of the authors that the expression in male is significantly higher than in women, I was wondering whether DYNC1|1 would be still useful as a prognostic marker for women with LIHC, or if the overexpression of this gene will remain significant in this comparison. Please, include these comparisons in this section and their statistics.

8. Line 133. Please, divide de survival rate also by gender in high and low DYNC1|1. Add the results in the text and the related graphics in supplementary information.

9. Line 153 to line 180. This text seems like an introduction and summary of the authors’ work. Just a reminder that this is the “Discussion” section. I suggest beginning with the discussion in line 181, maybe including a different first sentence that complete the paragraph. The previous discarded text could be included in the short introduction of de manuscript.

10. I should recommend authors to be more specific in some comments throughout the text because some of them like “its efficacy is still not optimistic” or “easy recurrence” do not sound very scientific. For avoiding being vague in the expression of this information, please provide previous statistics in related works supporting these comments.

11. In the discussion there is some information missing. Please provide more detail about the results in other cancers with DYNC1|1 (the sense of the deregulation, possible role on these cancers, etc) and add a proper comparison with the results obtained in this work.

12. Line 193. I am not totally sure what authors mean with the term “multi-cell cycle” and its regulation by DYNC1|1. Please, explain this concept.

Minor comments:

- Please, describe the abbreviations before including them in the text, i.e. like PPI (line 76).

- Line 80: Please change the sentence: “We use MCC, DMNC…” and substitute for “We used 11 algorithms of the Cyto-Hubba software to screen out the hub gene: MCC, DMNC, MNC…” instead, for expressing the methods a clearer way.

Reviewer #2: In this manuscript, the authors retrospectively analyzed the available TCGA data for LIHC to identify genes and pathways that are dysregulated in cancer compared to normal adjacent tissue. In their analyses, they identify DYNC1I1 as a hub gene in LIHC and a potential prognostic factor.

The analyses are well performed, however there are a few issues that should be solved before publications.

In general, I would highly recommend that they increase the font size of all the writing in the figures, most of which is currently not readable in the printed size.

Some more details would be appreciated in the figure legends. For example: specify what are the blue dots in figure 1B and the red dots in figure 1D. The legend of figure 3 has not details at all, it only indicates the tools that were used. In Figure 4A is the red line indicating the position of the gene? Please specify these details.

Cyto-Hubba identified DYNC1I1 as a hub gene. Was this the only gene at the intersection of the 11 lists? Please specify if this is the case.

Since they show that DYNC1I1 expression is higher in male patients (Figure 4D), it would be interesting to stratify the survival by sex and determine whether the prognostic significance is different in male vs female patients.

The first part of the discussion is more or less a repetition of the introduction and could be eliminated or incorporated in the introduction.

In the discussion they mention that DYNC1I1 is an independent prognostic factor, however, this is not fully demonstrated, because no multivariate analysis of the survival data is performed. Without this analysis, they should remove “independent”.

Finally, I would suggest that the authors adhere the general practice of writing in the past tense, if possible.

6. PLOS authors have the option to publish the peer review history of their article (what does this mean?). If published, this will include your full peer review and any attached files.

Reviewer #1: No

Reviewer #2: No

---

## [Author Response · Author response to Decision Letter 0]

15 Jul 2021

Response Letter

Dear Editor-in-Chief,

Thank you very much for your letter and advice. We have revised the manuscript, and would like to re-submit it for your consideration. We have addressed the comments raised by the reviewers and editor, and the amendments are highlighted in red in the revised manuscript. The responses to the editor' and reviewers' comments are listed below this letter.

We hope that the revised version of the manuscript is now acceptable for publication in your journal.

Looking forward to hearing from you soon.

With kind regards,

Yours sincerely,

Yan Liu Dr.

Puren Hospital Affiliated to Wuhan University of Science and Technology, Wuhan, Hubei, 430081, China. 

Tel/Fax: 027-86367776

Email: liuyan@wust.edu.cn

We would like to express our sincere thanks to the editor and reviewers for the constructive and positive comments.

Replies to Editor 

In the analysis it is stated that 371 samples of LIHC and adjacent normal tissue are used. However, the total number of "normal" samples is 50, so I would like to know if in a paired analysis (among the 50 paired samples) DYNC1I1 still appears as a prominent gene.

Answer: 

Thank you very much for your question. We extracted 50 pairs of matched patients and analyzed the data. The results show that DYNC1I1 is still a significant prominent gene.

On the other hand, the text included in the differential expression analysis is not very detailed. Has any kind of filter been done to eliminate low expressed genes ? with what cpm value ? The P value is adjusted ? with which method ?

Answer: 

Thank you for your question. We use the R package "Deseq2" to process the counts data of the transcriptome, and use the code "dds <- dds[rowSums(counts(dds))>1,]" to delete low-expressed genes. The CPM value should be applied to the "edgeR" package, but this R package was not used in our analysis. We use the R package "Deseq2" to adjust the P value. The code is 

"dds <-DESeqDataSetFromMatrix(countData=exprSet, colData=metadata, design=~sample,tidy=TRUE)

dds <- dds[rowSums(counts(dds))>1,]

dds <- DESeq(dds,parallel = T)". 

And our subsequent analysis is based on the Padj value.

Replies to Reviewer 1 

Reviewer #1: The present work describes the discovery of a novel prognostic biomarker for liver hepatocellular carcinoma patients. Using TCGA available RNA-seq and clinical patients’ data and applying bioinformatics approaches, authors unveil DYNC1|1 as the most significantly deregulated gene in a series of LIHC patients. I would like to address the following aspects related to this manuscript:

Major comments:

1.The main figures of the paper should represent the results described in the text. The figures selected do not have enough quality to be evaluated and the description in the legend is not well described. Please, add high quality figure that could be evaluated, a main title and a complete description of the plots and graphics represented, included the meaning of colour keys (colours should be correlating between figures) and symbols.

Answer: 

Thank you for your suggestion, we have updated the clarity of the figures and changed the description of the figures.

2.Line 102-103. Please, include a table (in supplementary data) with the 20 most significant GO terms described in the text and 9 KEGG significant pathways with their score. Provide plots with more quality.

Answer: 

Thank you for your suggestion, we have added the GO and KEGG data tables in the supplementary materials.

3.Line 119. Please provide a table (supplementary data) with the top ten genes obtained with the 11 different algorithms and their scores. Is only DYNC1|1 in the intersection of these algorithms? If not, include a ranking of the top ten (maximum) and their scores.

Answer: 

Thank you for your suggestion. There is only DYNC1I1 in the intersection of TOP100, so we provide the scores of the top ten genes and the scores of all genes in the supplementary materials. The file name is " Cyto-Hubba _score".

4.It would be interesting if authors could provide a table with the interacting genes belonging to the same hub than DYNC1|1. It could complete the information and add some valuable insight about the possible functionality of this alteration in liver cancer (main text).

Answer: 

Thank you for your suggestion. We have added a list of proteins that interact with DYNC1I1 in the main text.

5.Line 124. In the characterization of DYNC1|1 there is a lack of description. Authors showed in a figure the gene location and protein structure, but they did not add much detail in the text. Please, complete the description (family of the gene, which kind of protein codifies…), features of the structure and potential functionality of the gene (if it was previously described).

Answer: 

Thanks for your suggestion, we have added a related description in the article.

6.Line 126. Authors described the significant difference of DYNC1|1 expression between LIHC and normal adjacent tissue, but they did not include the statistical value for this comparison in the text. The same was done when they compared male vs female in LIHC. Please, include all these statistics, and add in the “Methods” sections how these statistics were performed.

Answer: 

Thank you for your suggestion. We have added the P value data to the main text, and described the use of the R package "DESeq2" to analyze the data in the method.

7.In line with the previous comments and comparisons performed, the individual comparisons “LIHC vs control” in male and women independently were missed, and due to the observation of the authors that the expression in male is significantly higher than in women, I was wondering whether DYNC1|1 would be still useful as a prognostic marker for women with LIHC, or if the overexpression of this gene will remain significant in this comparison. Please, include these comparisons in this section and their statistics.

Answer: 

Thank you for your valuable comments. Our research found that the expression of DYNC1I1 in men is higher than that in women, and it can predict the prognosis very well in men, but not in women. We have added this part to the main text.

8.Line 133. Please, divide de survival rate also by gender in high and low DYNC1|1. Add the results in the text and the related graphics in supplementary information.

Answer: 

Thank you for your comments. We have added this part to the main text.

9.Line 153 to line 180. This text seems like an introduction and summary of the authors’ work. Just a reminder that this is the “Discussion” section. I suggest beginning with the discussion in line 181, maybe including a different first sentence that complete the paragraph. The previous discarded text could be included in the short introduction of de manuscript.

Answer: 

Thank you for your comments, we have made changes to this part of the content.

10.I should recommend authors to be more specific in some comments throughout the text because some of them like “its efficacy is still not optimistic” or “easy recurrence” do not sound very scientific. For avoiding being vague in the expression of this information, please provide previous statistics in related works supporting these comments.

Answer: 

Thanks for your suggestions, we have modified these descriptions accordingly.

11.In the discussion there is some information missing. Please provide more detail about the results in other cancers with DYNC1|1 (the sense of the deregulation, possible role on these cancers, etc) and add a proper comparison with the results obtained in this work.

Answer: 

Thank you for your suggestions. We have added and modified relevant content according to your suggestions.

12.Line 193. I am not totally sure what authors mean with the term “multi-cell cycle” and its regulation by DYNC1|1. Please, explain this concept.

Answer: 

Thank you for your correction. Due to our negligence, we can see that DYNC1I1 regulates the cell cycle in KEGG, which we have corrected in the article.

Minor comments:

- Please, describe the abbreviations before including them in the text, i.e. like PPI (line 76).

Answer: 

Thanks for your suggestion, we have revised the acronyms.

- Line 80: Please change the sentence: “We use MCC, DMNC…” and substitute for “We used 11 algorithms of the Cyto-Hubba software to screen out the hub gene: MCC, DMNC, MNC…” instead, for expressing the methods a clearer way.

Answer: 

Thanks for your suggestion, we have made changes.

Reviewer #2: In this manuscript, the authors retrospectively analyzed the available TCGA data for LIHC to identify genes and pathways that are dysregulated in cancer compared to normal adjacent tissue. In their analyses, they identify DYNC1I1 as a hub gene in LIHC and a potential prognostic factor.

The analyses are well performed, however there are a few issues that should be solved before publications.

In general, I would highly recommend that they increase the font size of all the writing in the figures, most of which is currently not readable in the printed size.

Answer: 

Thanks for your suggestion, we have updated the figures.

Some more details would be appreciated in the figure legends. For example: specify what are the blue dots in figure 1B and the red dots in figure 1D. The legend of figure 3 has not details at all, it only indicates the tools that were used. In Figure 4A is the red line indicating the position of the gene? Please specify these details.

Answer: 

Thank you for your suggestions. We have added and modified relevant content according to your suggestions.

Cyto-Hubba identified DYNC1I1 as a hub gene. Was this the only gene at the intersection of the 11 lists? Please specify if this is the case.

Answer: 

Yes, this is the only gene in the TOP100 among the 11 algorithms.

Since they show that DYNC1I1 expression is higher in male patients (Figure 4D), it would be interesting to stratify the survival by sex and determine whether the prognostic significance is different in male vs female patients.

Answer: 

Thank you for your suggestion. DYNC1I1 has different effects on the survival of men and women, and we have added this content in the main text.

The first part of the discussion is more or less a repetition of the introduction and could be eliminated or incorporated in the introduction.

Answer: 

Thanks for your suggestion, we revised the first part of the discussion.

In the discussion they mention that DYNC1I1 is an independent prognostic factor, however, this is not fully demonstrated, because no multivariate analysis of the survival data is performed. Without this analysis, they should remove “independent”.

Answer: 

Thank you for your suggestion. We changed the "independent prognosis" to "prognosis".

Finally, I would suggest that the authors adhere the general practice of writing in the past tense, if possible.

Answer: 

Thanks for your suggestion, we will ask native English speakers to help us modify the grammar of the article.

---

## [Decision Letter · Decision Letter 1]

9 Aug 2021

PONE-D-21-13904R1

Bioinformatics analysis identifies DYNC1I1 as prognosis marker in male patients with Liver hepatocellular carcinoma

PLOS ONE

Dear Dr. Liu,

Thank you for submitting your manuscript to PLOS ONE. After careful consideration, we feel that it has merit but does not fully meet PLOS ONE’s publication criteria as it currently stands. Therefore, we invite you to submit a revised version of the manuscript that addresses the points raised during the review process.

You will see that while the reviewers are persuaded of the importance of your study and agree with your revisions, reviewer 1 has suggested several minor changes to the text. In addition I have several comments below that require clarification.

We look forward to receiving your revised manuscript.

Kind regards,

Katherine James, Ph.D.

Academic Editor

PLOS ONE

Journal Requirements:

Additional Editor Comments:

PlosOne requires methods to be described in sufficient detail for another researcher to reproduce the experiments described and currently your analyses would not be reproducible from the provided methodology. I have several minor comments that require clarification:

Please clarify package names and versions for all R packages used.

Please also clarify parameters used in all software. If defaults were used please state “using default parameters” or similar in the text.

You have clarified processing of the RNAseq data in the response to review: “use the code "dds <- dds[rowSums(counts(dds))>1,]" to delete low-expressed genes”. This information should be in the methods.

Please ensure all methodology is covered, for instance the DAVID database is mentioned in the results section but not in the methods.

Additionally, what is the source of the structural data in figure 4?

Please add appropriate references for all software and databases used.

The STRING database contains multiple types of interaction from several sources, many of which are functional interactions rather than physical PPI eg co-expression data. Please detail the sources chosen and score threshold applied. If functional data are included in the network, please change the text accordingly as this is not a PPI network.

At line 213 please change “gender” to “sex” to clarify this is a biological classification.

Reviewers' comments:

Reviewer's Responses to Questions

**Comments to the Author**

1. If the authors have adequately addressed your comments raised in a previous round of review and you feel that this manuscript is now acceptable for publication, you may indicate that here to bypass the “Comments to the Author” section, enter your conflict of interest statement in the “Confidential to Editor” section, and submit your "Accept" recommendation.

Reviewer #1: All comments have been addressed

Reviewer #2: All comments have been addressed

2. Is the manuscript technically sound, and do the data support the conclusions?

Reviewer #1: Yes

Reviewer #2: Yes

3. Has the statistical analysis been performed appropriately and rigorously?

Reviewer #1: Yes

Reviewer #2: Yes

4. Have the authors made all data underlying the findings in their manuscript fully available?

Reviewer #1: Yes

Reviewer #2: Yes

5. Is the manuscript presented in an intelligible fashion and written in standard English?

Reviewer #1: Yes

Reviewer #2: Yes

6. Review Comments to the Author

Reviewer #1: The authors have completed the requests and added some valuable information and discussion about the different aspects pointed out during the first revision, improving considerably the quality of the manuscript. I am happy with the result of these changes. However, there is still some minor remaining aspects that I would like to point out before the manuscript will be ready for publication:

Minor comments:

1. Line 69. There are still some misspelling words. Please, substitute “considerd” for “considered” and “logFC <2.5 are significant” for “logFC >2.5 as significant”.

2. Lines 108-109. Please, include the information “using the R package “clusterprofiler” in Methods’ section, “Functional enrichment analysis” (line 73) where it corresponds. This information should be there and not in the Results’ section. Same happened with the sentence: “GO function annotation of the interacting proteins was performed using the R package” (line 170-171), move that information to the same section.

3. Lines 142, 145, 148, 154. Please, substitute “than that” only for “than”.

4. Line 147. Write “and the results showed that…” instead of “and the resulted show that…”.

5. Lines 157-158. Please, insert here that “These results indicated that high expression of DYNC1|1…”.

6. Lines 186, 188, 189. Please, eliminate in the indicated lines the word “rate” after “recurrence”, “recurrence” and “survival” consecutively.

7. Lines 209, 211. Add “LIHC in male patients” instead “LIHC male patients”.

8. Line 212. Same here, add “LIHC for male patients”.

9. Line 213. “we will classify patients’ samples by gender…”.

Reviewer #2: (No Response)

7. PLOS authors have the option to publish the peer review history of their article (what does this mean?). If published, this will include your full peer review and any attached files.

**Do you want your identity to be public for this peer review? **For information about this choice, including consent withdrawal, please see our Privacy Policy.

Reviewer #1: No

Reviewer #2: No

---

## [Author Response · Author response to Decision Letter 1]

13 Aug 2021

Response Letter

Dear Editor-in-Chief,

Thank you very much for your letter and advice. We have revised the manuscript, and would like to re-submit it for your consideration. We have addressed the comments raised by the reviewers and editor, and the amendments are highlighted in red in the revised manuscript. The responses to the editor' and reviewers' comments are listed below this letter.

We hope that the revised version of the manuscript is now acceptable for publication in your journal.

Looking forward to hearing from you soon.

With kind regards,

Yours sincerely,

Yan Liu Dr.

Puren Hospital Affiliated to Wuhan University of Science and Technology, Wuhan, Hubei, 430081, China. 

Tel/Fax: 027-86367776

Email: liuyan@wust.edu.cn

We would like to express our sincere thanks to the editor and reviewers for the constructive and positive comments.

Replies to Journal Requirements

Answer: Thank you for your suggestions. We have revised and updated the references. 

Replies to Editor 

Please clarify package names and versions for all R packages used.

Answer: Thank you for your valuable comments. We updated the package name and version of all the R package. 

Please also clarify parameters used in all software. If defaults were used please state “using default parameters” or similar in the text.

Answer: Thank you for your comment, we revised the relevant content.

You have clarified processing of the RNAseq data in the response to review: “use the code "dds <- dds[rowSums(counts(dds))>1,]" to delete low-expressed genes”. This information should be in the methods.

Answer: Thank you for your suggestions, we revised the relevant content.

Please ensure all methodology is covered, for instance the DAVID database is mentioned in the results section but not in the methods.

Answer: Thank you for your comment. We revised the relevant content.

Additionally, what is the source of the structural data in figure 4?

Answer: Thanks for your comment. Data was obtained from the online database Swiss-Model (https://swissmodel.expasy.org/repository/uniprot/O14576?csm=55A6FF971E632DA0), and we revised the relevant content in text.

Please add appropriate references for all software and databases used.

Answer: Thank you for your suggestions. We have added relevant references.

The STRING database contains multiple types of interaction from several sources, many of which are functional interactions rather than physical PPI eg co-expression data. Please detail the sources chosen and score threshold applied. If functional data are included in the network, please change the text accordingly as this is not a PPI network.

Answer: Thanks for your suggestion. Homo sapiens was selected for STRING analysis, and the detailed scoring value was added to the supplementary materials.

At line 213 please change “gender” to “sex” to clarify this is a biological classification.

Answer: Thank you for your comment, we have made changes to this part of the content. 

Replies to Reviewer 1 

Reviewer #1: The authors have completed the requests and added some valuable information and discussion about the different aspects pointed out during the first revision, improving considerably the quality of the manuscript. I am happy with the result of these changes. However, there is still some minor remaining aspects that I would like to point out before the manuscript will be ready for publication:

Minor comments:

1.Line 69. There are still some misspelling words. Please, substitute “considerd” for “considered” and “logFC <2.5 are significant” for “logFC >2.5 as significant”.

Answer: Thanks for your suggestions, we have made changes.

2.Lines 108-109. Please, include the information “using the R package “clusterprofiler” in Methods’ section, “Functional enrichment analysis” (line 73) where it corresponds. This information should be there and not in the Results’ section. Same happened with the sentence: “GO function annotation of the interacting proteins was performed using the R package” (line 170-171), move that information to the same section.

Answer: Thank you for your valuable comments. We have made corresponding modifications in the article. 

3.Lines 142, 145, 148, 154. Please, substitute “than that” only for “than”.

Answer: Thank you for your suggestions. We changed the "than that" to "than".

4.Line 147. Write “and the results showed that…” instead of “and the resulted show that…”.

Answer: Thanks for your suggestion, we have made changes.

5.Lines 157-158. Please, insert here that “These results indicated that high expression of DYNC1|1…”.

Answer: Thanks for your suggestion, we have modified these descriptions accordingly.

6.Lines 186, 188, 189. Please, eliminate in the indicated lines the word “rate” after “recurrence”, “recurrence” and “survival” consecutively.

Answer: Thanks for your suggestions, we have made changes.

7.Lines 209, 211. Add “LIHC in male patients” instead “LIHC male patients”.

Answer: Thanks for your suggestions, we changed the "LIHC male patients" to "LIHC in male patients".

8.Line 212. Same here, add “LIHC for male patients”.

Answer: Thank you for your suggestion, we have made changes to this part of the content. 

9. Line 213. “we will classify patients’ samples by gender…”.

Answer: Thanks for your suggestion, we have made changes.

---

## [Editor Report · Decision Letter 2]

6 Oct 2021

Bioinformatics analysis identifies DYNC1I1 as prognosis marker in male patients with Liver hepatocellular carcinoma

PONE-D-21-13904R2

Dear Dr. Liu,

We’re pleased to inform you that your manuscript has been judged scientifically suitable for publication and will be formally accepted for publication once it meets all outstanding technical requirements.

Kind regards,

Katherine James, Ph.D.

Academic Editor

PLOS ONE
---

## [Editor Report · Acceptance letter]

14 Oct 2021

PONE-D-21-13904R2 

Bioinformatics analysis identifies DYNC1I1 as prognosis marker in male patients with Liver hepatocellular carcinoma 

Dear Dr. Liu:

I'm pleased to inform you that your manuscript has been deemed suitable for publication in PLOS ONE. Congratulations! Your manuscript is now with our production department. 

Kind regards, 

on behalf of

Dr. Katherine James 

Academic Editor

PLOS ONE